# Diamond Nanoparticles Downregulate Expression of *CycD* and *CycE* in Glioma Cells

**DOI:** 10.3390/molecules24081549

**Published:** 2019-04-19

**Authors:** Marta Grodzik, Jaroslaw Szczepaniak, Barbara Strojny-Cieslak, Anna Hotowy, Mateusz Wierzbicki, Slawomir Jaworski, Marta Kutwin, Emilia Soltan, Tomasz Mandat, Aneta Lewicka, Andre Chwalibog

**Affiliations:** 1Division of Nanobiotechnology, Faculty of Animal Sciences, Warsaw University of Life Sciences, Ciszewskiego 8, 02-786 Warsaw, Poland; jaroslaw_szczepaniak@sggw.pl (J.S.); barbara_strojny@sggw.pl (B.S.-C.); anna_hotowy@sggw.pl (A.H.); mateusz_wierzbicki@sggw.pl (M.W.); slawomir_jaworski@sggw.pl (S.J.); marta_kutwin@sggw.pl (M.K.); 2Department of Neurosurgery, Oncology Center- Maria Sklodowska Curie Memorial, Warsaw, Roentgena 5, 02-781 Warsaw, Poland; emiliasoltan@gmail.com (E.S.); tomaszmandat@yahoo.com (T.M.); 3Laboratory of Epidemiology, Military Institute of Hygiene and Epidemiology, Kozielska 4, 01-163 Warsaw, Poland; anet.lewicka@gmail.com; 4Department of Veterinary and Animal Sciences, University of Copenhagen, Groennegaardsvej 3, 1870 Frederiksberg, Denmark; ach@sund.ku.dk

**Keywords:** diamond nanoparticles, glioblastoma, proliferation, cell cycle, cancer

## Abstract

Our previous studies have shown that diamond nanoparticles (NDs) exhibited antiangiogenic and proapoptotic properties in vitro in glioblastoma multiforme (GBM) cells and in tumors in vivo. Moreover, NDs inhibited adhesion, leading to the suppression of migration and invasion of GBM. In the present study, we hypothesized that the NDs might also inhibit proliferation and cell cycle in glioma cells. Experiments were performed in vitro with the U87 and U118 lines of GBM cells, and for comparison, the Hs5 line of stromal cells (normal cells) after 24 h and 72 h of treatment. The analyses included cell morphology, cell death, viability, and cell cycle analysis, double timing assay, and gene expression (*Rb, E2F1, CycA, CycB, CycD, CycE, PTEN, Ki-67*). After 72 h of ND treatment, the expression level of *Rb, CycD*, and *CycE* in the U118 cells, and *E2F1*, *CycD*, and *CycE* in the U87 cells were significantly lower in comparison to those in the control group. We observed that decreased expression of cyclins inhibited the G1/S phase transition, arresting the cell cycle in the G0/G1 phase in glioma cells. The NDs did not affect the cell cycle as well as *PTEN* and *Ki-67* expression in normal cells (Hs5), although it can be assumed that the NDs reduced proliferation and altered the cell cycle in fast dividing cells.

## 1. Introduction

Diamond nanoparticles (NDs) are one of the allotropic forms of carbon in addition to others like fullerene, nanotubes, graphene or graphite. As in the case of other carbon materials, their biological and chemical properties depend on the synthesis method and post-synthesis modifications. Diamond nanoparticles produced by the detonation (non-chemical) method [1] have several common features. Their particle size is approximately 5 nm, and the core is a diamond (carbon of sp^3^ hybridization) coated with a layer of amorphous carbon (sp^2^ hybridized). Of the oxygen groups resulting from the purification of the powder in an oxygen atmosphere, the hydroxyl, carboxylic, and epoxide groups are predominantly present on the surface of the nanoparticles [2].

Glioblastoma multiforme (GBM) is one of the most malicious types of brain tumors, which qualifies for IV World Health Organization status, the highest class of malignancy. Due to such factors as its location, rapid growth rate, cell migration along the nerves and blood vessels, and genetic instability, there is still a lack of effective therapies to combat GBM [3].

The current standard therapy is maximal surgical resection followed by radiotherapy and chemotherapy with temozolamide [4]. The main problem in chemotherapy is the presence of the blood–brain barrier (BBB), which blocks toxins, as well as many essential drugs, from reaching brain tissue. Nanotechnology and mathematical modeling of nanoparticles can help the delivery of active compounds (for example delphinidin) to brain tissue [5], while other authors have used polymeric nanoparticles with Paclitaxel [6], polymeric micelles with doxorubicin [7], and liposomes with siRNA [8] for convection-enhanced delivery. In the current article, we focus on nanoparticles not as vehicles for delivery, but as active agents with anticancer properties. The medical use of NDs is probable because of their low cytotoxicity and high biological activity [9]. Several authors have reported a lack of toxic effects that manifest in the forms of inflammation and activation of macrophages [10], and symptoms of systemic toxicity after ND treatment in rats [11,12], but have also noted uptake and intracellular accumulation of NDs [13], activation of cell death (both apoptotic and necrotic) [14], and redox homeostasis disorders [15]. Diamond nanoparticles were also tested for their application in cancer chemotherapy. Yu et al. [16] synthesized diamond-based nanoparticles which can penetrate cell membranes in vitro and in vivo. On the other hand, NDs functionalized by Epirubicin improved impairment of secondary tumor formation in vivo [17].

In the case of GBM, NDs have been observed to reduce the vascularization of GBM tumors grown on the chick embryo’s chorioallantoic membrane (CAM) by lowering the expression of the vascular endothelial growth factor (*VEGF*) and fibroblast growth factor 2 *FGF2* genes, and also to lower the tumor mass and volume [18,19]. In in vitro studies, it has been observed that NDs inhibit the adhesion of U87 and U118 cells, thereby leading to suppression of migration and invasiveness, through modulation of the epidermal growth factor receptor/protein kinase-B/mammalian target of rapamycin (EGFR/AKT/mTOR) pathway as well as by decreasing the expression of β-catenin [20]. β-catenin is a multifunctional protein involved in cell–cell adhesion, induction of cell proliferation in a variety of tumors, and regulation of the cell cycle [21].

In the light of altered activity of the EGFR/AKT/mTOR pathway and decreased β-catenin expression in the nucleus, we hypothesized that NDs can decrease proliferation by arresting the cell cycle of glioblastoma cells. As reduced proliferation can be caused by the arrest of the cell cycle in different stages, we decided to investigate the genes related to G1/S phase transition, namely, retinoblastoma protein (*Rb*), transcription factor E2F1 (*E2F1*), cyclin D (*CycD*), cyclin E (*CycE*), and marker of proliferation KI-67 (*Ki-67*) as some of the possible points of ND action. To examine the cell cycle, we applied the propidium iodide (PI) flow cytometry assay and evaluation of the level of gene expression following the incubation of U87 and U118 cells with NDs for 24 and 72 h. To prove that the effect is characteristic for cancer cells, complementary experiments with normal cells (Hs5) were performed in which the viability, proliferation, and cell cycle as well as the expression of proliferating cell nuclear antigen (*PCNA*) and *Ki-67* genes were assessed.

## 2. Results and Discussion

### 2.1. Characterization of NDs and Analysis of Cell Viability

The transmission electron microscopy (TEM) image, X-ray diffraction (XRD) diagram, results of the zeta potential and dynamic light scattering (DLS) of NDs are presented in Figure 1. The TEM analyses were used to examine the morphology of the nanoparticles. Additionally, DLS analysis was performed to determine the average hydrodynamic diameter of NDs. The zeta potential was analyzed to characterize the surface charges and the stability of the ND suspensions [22]. The NDs were 4–5 nm in diameter and spherical in shape. The XRD analysis showed three reflections and the position and width of these reflections corresponded to the lattice parameters characteristic of diamond nanoparticles [23]. The zeta potential of the hydrocolloid NDs was +28.9 with a standard deviation ±6.64 which indicates an incipient instability. The size distribution shows the presence of three fractions of particles with dimensions of 4, 5, and 20 nm. The biggest fractions were probably the result of agglomeration of the smaller ones. The surface functional groups of NDs have been described in our previous publication [24]. Kurantowicz et al. [24] obtained Fourier-Transform Infrared Spectroscopy (FTIR) spectra for NDs. The most intense band at 3430–3444 cm^−1^ point to the O−H stretching vibrations of hydroxyl groups in adsorbed water molecules, structural OH groups, and carboxylic acids. Peaks at 1720–1757 cm^−1^ are characteristic for C=O stretching vibrations from carbonyl and carboxylic groups and at 1239–1261 cm^−1^ caused by C−O−C stretching vibrations from epoxy-functional groups.

The physicochemical parameters of NDs were similar to those previously described [25,26,27,28].

In order to evaluate the ND toxicity in GBM (U87, U118) and normal (Hs5) cells, the cell morphology and survival rate were examined. The images of cells treated with 5 and 50 μg/mL ND concentrations are shown in Figure 2. After 24 h, when compared to the control, the treated U87 and U118 cells showed no changes in morphology but were found to be less dense at all concentrations. However, when the cells were incubated with 50 μg/mL of ND for 72 h, they exhibited decreased cell density and morphological changes such as the formation of round-shaped cells, cell shrinkage, and spherical cellular protrusions formation. The changes were more visible in the U87 cells than in the U118 cells and predominantly in the wells with few cells but not in all of them. Agglomerates of NDs were not noticeable, but GBM cells treated with 50 μg/mL of NDs acquired a light-brown hue. To determine the differences in ND interaction between cancer cells and normal cells, the morphology of Hs5 cells after 5 and 50 μg/mL ND treatment was also evaluated. After incubation with 50 μg/mL of ND for 72 h, the HS5 cells were spindle-shaped with nanoparticles inside and agglomerates of NDs visible outside of cells.

The viability of the U87, U118, and Hs5 cells was examined using the 3-(4, 5-dimethylthiazolyl-2)-2, 5-diphenyltetrazolium bromide (MTT) assay (Figure 3). The treatment with 50 μg/mL concentration of ND for 24 and 72 h significantly reduced the viability of the U87 cells by 20 and 38% and that of the U118 cells by 16 and 30%. In Hs5 cells after incubation with 50 μg/mL concentration of ND for 24 and 72 h, viability was reduced by 24 and 19%, respectively.

The percentage of apoptotic cells (early and late in total) (Figure 4) increased to 14.5 and 25.2% after treating the U87 cells with 50 μg/mL of ND for 24 and 72 h, respectively; in the case of U118 cells, the percent increased to 9.5 and 20.6% after 24 and 72 h, respectively. These results demonstrate the ability of NDs to induce apoptosis in U87 and U118 cells in a time-dependent manner. In Hs5 cells, apoptosis/necrosis was induced with 50 μg/mL of NDs for 72 h; however, the percentage of dead cells was lower than in the U87 and U118 groups.

In general, NDs are characterized by low toxicity [28,29] and high biocompatibility [30]. Schrand et al. [31] examined the 24 h effect of NDs on a variety of cells, including neuroblastomas, macrophages, keratinocytes, and PC-12 cells using the MTT assay and by ultrastructural cell morphology analysis. No significant changes were observed; however, the neuroblastoma cells lost their normal morphology after incubation with 100 μg/mL of NDs. In our experiments, we observed some alterations in the glioblastoma cell morphology in both cell lines. It was not visible for all the cells in wells, but changes that have occurred were characteristic for the induction of apoptosis. Moreover, Hs5 cells were less sensitive to NDs than GBM cells, only after treatment with 50 μg/mL of ND was observed lower viability than in control group. Stojny [32] compared the effect of NDs on fibroblast and HepG2 cell lines. In that experiment, the normal cell line was more sensitive to the treatment than the cancer cell line. Similarly, in a comparative study of the cytotoxicity of detonation nanodiamonds on the osteosarcoma cell line (MG-63) and the primary mesenchymal stem cells (rMSCs), cancer cells were more resistant to NDs than stem cells [27]. Apoptosis, a form of programmed cell death, is the most expected form of death not only in cancer cells [33] but also in normal cells, especially during cell differentiation. It was observed that ND treatment of human umbilical vein endothelial cells (HUVECs) could induce apoptosis and/or necrosis, depending on the treatment time and concentration of NDs [14]. Furthermore, in macrophages grown in the presence of NDs (50 μg/mL), the incidence of apoptosis increased by approximately 22% [34]. Presented experiments on U87 and U118 GBM cells also confirmed the proapoptotic activity of NDs. Keremidarska et al. [27] summarized their experiment that the response of a given cell line is likely to vary considerably, and that even different kinds of cancer cells can probably have diverse reactions. Cell viability in response to ND treatment should not be generalized for all cancer or non-cancer cells.

### 2.2. Analysis of Proliferation

All cells, including cancer cells, have a characteristic balance between apoptosis and proliferation [34]. The disturbance of one of these processes is reflected as the changes in the other. The proliferation test demonstrated that both cancer cell lines after 72 h treatment with NDs at concentrations of 20 and 50 μg/mL significantly reduced cell proliferation by 40 and 45% in U87 cells and by 15 and 22% in the U118 cells, respectively (Figure 5). Moreover, proliferation decreased by 23 and 29% in the U87 cells treated for 72 h with 5 and 10 μg/mL ND, respectively. Inhibition of proliferation was observed in both cell lines after 72 h treatment with 50 μg/mL of NDs. However, proliferation of the normal Hs5 cells did not change.

The inhibition of proliferation can also be evaluated by the doubling time assay. The growth rate was measured after the 24 and 72 h ND treatments at concentrations of 5 and 50 μg/mL (Table 1). The doubling time was 32 and 19 h in the control of U87 and U118 cell lines, respectively. The doubling time for the U87 cells treated with 5 and 50 μg/mL of ND was 34 and 38 h, respectively, and those for the U118 cells were 19 and 23 h, respectively. Compared to the control, the 50 μg/mL treatment increased the doubling time of U87 cells by 7 h and that of U118 by 4 h. Doubling time of Hs5 was prolonged by 1 h in groups treated with ND compared to the control, from 71 to 72 h.

The expression of two key genes involved in the regulation of cell proliferation, namely *PCNA* and *Ki-67* were analyzed (Figure 6). At the ND concentration of 50 μg/mL, the expression of *Ki-67* significantly decreased in the U87 cells following 24 h of treatment, but decreased in both the U87 and U118 cells after 72 h of treatment in comparison to that of the control group. There was no effect on *PCNA* and *Ki-67* expression after ND treatment in Hs5 cells.

The expression of the human Ki-67 protein is strictly associated with cell proliferation. The fact that the Ki-67 protein is present during all active phases of the cell cycle (G_1_, S, G_2_, and mitosis) but is absent in the resting cells (G_0_) makes it an excellent marker for determining the so-called growth fraction of a given cell population [35]. We have previously demonstrated that the administration of 50 μg/mL NDs for 72 h inhibits the cell proliferation in the U87 line and is also associated with a decrease in *PCNA* expression [20]. However, it did not affect the U118 cells in the present experiment. Differences in the responses of these lines may result from the different genetic profiles of the U87 and U118 cells. Both cell lines carry mutations in *PTEN*, whereas U118 also exhibits the *p53* mutation [36]. The *Ki-67* expression was not lowered in cells carrying mutations in the *PTEN* and *p53* genes [37]. Thus, it can be suspected that p53 takes part in the mechanism of ND action on GBM cells. However, decreased proliferation, elongated doubling time, and decreased expression of *Ki-67* in U87 and U118 cells after 72 h of incubation with NDs suggest an arrest of the cell cycle at the G_0_/G_1_ phase. Normal cells Hs5 were non-sensitive to ND treatment, and none of the tested genes had changes in their level of expression.

### 2.3. Analysis of Cell Cycle

To deepen the previous findings of the effect of NDs on the cell cycle, we analyzed the PI/DN-ase cell cycle assay and by measuring the mRNA expression level of select genes (*CycA*, *CycB*, *CycD*, *CycE*, *Rb, E2F1*). The cell cycle consists of four distinct phases, namely, the G_1_ phase, S phase (synthesis), G_2_ phase (collectively known as the interphases), and the M phase (mitosis). The cell grows at a steady rate throughout the interphases, with most dividing cells doubling in size between one mitosis and the next. The M phase of the cycle corresponds to mitosis, which is usually followed by cytokinesis. This phase is followed by the G_1_ phase (gap 1), which corresponds to the interval between mitosis and the initiation of DNA replication. During G_1_, the cell is metabolically active and continuously grows but does not replicate its DNA. G_1_ is followed by the S phase (synthesis), during which DNA replication takes place. The completion of DNA synthesis is followed by the G_2_ phase, during which cell growth continues and proteins are synthesized in preparation for mitosis [38].

All cell lines were subjected to 24 and 72 h treatments with ND concentrations of 5 and 50 μg/mL. As shown in Figure 7, the G_0_/G_1_ phase of cells was arrested in both the U87 and U118 cells treated with 50 μg/mL NDs for 72 h. The number of G_2_/M phase cells was significantly lower in the U87 and U118 cells treated with 50 μg/mL ND for 72 h in comparison to that of the control. However, no effect was observed on the Hs5 cell cycle.

The induction of the G_0_/G_1_-phase arrest caused by NDs was further confirmed by qPCR analysis (Figure 8). The expression levels of *Rb* (retinoblastoma tumor suppressor), *CycD*, *CycE*, and *E2F1*, *CycD*, *CycE* in the U118 cells and in the U87 cells, respectively, treated with 50 μg/mL ND for 72 h were significantly lower in comparison to the those in the control group.

This is the first report suggesting possibility of stopping the cell cycle in GBM cells between mitosis and the replication of DNA using NDs. The results obtained from qPCR were consistent with those obtained from the flow cytometric analysis of the cell cycle, thereby confirming the inhibition of the G_0_/G_1_ phase by downregulation of the expression of the cyclins, *CycD* and *CycE*. Cyclin D and cyclin E are the two major classes of G_1_ cyclins. These proteins are key regulators of the cell cycle progression, capable of interacting and activating specific cyclin-dependent kinases (CDKs). Cyclin D interacts with either CDK4 or CDK6, and cyclin E partners with CDK2. Cyclin D interacts with either CDK4 or CDK6, and cyclin E couples with CDK2. Cyclin D, connected with either CDK4 or CDK6, phosphorylates the retinoblastoma tumor suppressor, Rb, existing as a complex with E2F to generate hypophosphorylated Rb and free E2F. The E2F family of transcription factors are critical for progression into the S phase [39]. Thus, the decreased expression of these cyclins blocks the transition to the S phase (Figure 9).

Analyzing potential molecular pathways leading to cell cycle arrest in the G_0_ phase, ND interaction with tyrosine kinase receptors appears probable. It was proved that ND affected the EGF receptor [20] and the VEGF receptor [40]. Presumably, NDs can modulate receptor autophosphorylation, and in the next order, Ras activity. Ras is responsible for cyclin D induction via mitogen-activated protein kinase (MAPK) and cell–cycle progression end, on the other hand, Ras inhibition slows down or stops cell division [41].

In the present experiment, there was actually no response from HS5 normal cells to applied ND concentrations ranging from 1–50 μg/mL. An important difference between Hs5 and glioma cell lines (U87 and U118) is overexpression of tyrosine kinase receptors [42]. Probably, NDs can inhibit autophosphorylation, and consequently, leads to normalization of proliferation of cancer cells.

## 3. Material and Methods

### 3.1. Nanomaterials

Diamond nanoparticles produced by the detonation method were purchased from SkySpring Nanomaterials (Houston, TX, USA). According to the manufacturer’s specification, they had a purity of >95% with a surface area of ~282 m^2^/g. The nanopowder was suspended in ultrapure water to prepare a 1.0 mg/mL stock solution. Immediately before exposure to cells, hydrocolloids of NDs were sonicated for 30 min and diluted to concentrations 5 and 50 μg/mL with supplemented Dulbecco’s Modified Eagle’s culture Medium (DMEM, Thermo Fisher Scientific, Waltham, MA, USA).

### 3.2. Visualization of Nanoparticles

The shape and size of the nanoparticles were inspected using the transmission electron microscope (TEM), JEM-1220 (JEOL, Tokyo, Japan) at 80 kV with a Morada 11-megapixel camera (Olympus Soft Imaging Solutions, Münster, Germany). An aliquot of 5 μL of 50 μg/mL ND sample was placed onto formvar-coated copper grids (Agar Scientific Ltd., Stansted, UK) and air dried prior to observation.

### 3.3. Zeta Potential

Zeta potential was measured using the Smoluchowski approximation. The hydrodynamic diameter was measured on the basis of dynamic light scattering (DLS). Both measurements were carried out using the Nano-ZS90 Zetasizer (Malvern Instruments, Malvern, UK).

### 3.4. Cells and Cell Cultures

The human GBM cell lines (U87 and U118) and normal stroma cell line (Hs5) were obtained from the American Type Culture Collection (Manassas, VA, USA) and cultured in Dulbecco’s Modified Eagle’s Medium (Thermo Fisher Scientific, MA, USA) with the addition of 10% fetal bovine serum (FBS, Thermo Fisher Scientific) and 1% penicillin and streptomycin (Thermo Fisher Scientific) at 37 °C in a humidified atmosphere of 5% CO_2_ inside the incubator, INCOMED153 (Memmert GmbH & Co, Germany).

### 3.5. Cell Morphology

The U87, U118, and Hs5 cell lines (5 × 10^4^ cells per well) were seed onto 6-well plates and incubated overnight. Then, they were treated with NDs at concentrations of 5 and 50 μg/mL. Cells cultured without the addition of nanoparticles were used as the control group. After 24 and 72 h of incubation, the changes in the morphology of the cells was investigated using a CKX 41 epifluorescent inverted microscope with a fluorescent filter (Olympus, Tokyo, Japan). The images were captured using a ProgRes^®^ c12 camera (Jenoptik, Jena, Germany).

### 3.6. Cell Viability Assays

The viability of U87, U118, and Hs5 cells was investigated using the colorimetric 3-(4,5-dimethylthiazol-2-yl)-2,5-diphenyltetrazolium bromide (MTT) assay. Both U87 and U118 cells were plated onto 96-well plates (5 × 10^3^ cells per well) and incubated overnight. Then, the medium was removed and replaced with a medium containing NDs at concentrations of 1, 5, 10, 20 or 50 μg/mL and incubated for 24 or 72 h. The control group received DMEM only. First, 15 μg of MTT (5 mg/mL) was added per well, and after 3 h a detergent (isopropanol with a drop of HCl) was added. A multi-well plate reader (Infinite^®^ 200 PRO) i-control™ Software (Tecan Group Ltd., Männedorf, Germany) was used for measuring the absorbance at 570 nm. Cell viability was expressed as the percentage of the control group viability, which was 100%, and plotted as the surviving fraction of cells relative to the no-treatment condition. Calculations were performed as described by Strojny et al. [32].

### 3.7. Apoptosis and Necrosis Assay

Annexin V and propidium iodide (PI) staining for the apoptosis/necrosis assay was performed using the Alexa Fluor^®^ 488 Annexin V/Dead Cell Apoptosis Kit (Thermo Fisher Scientific, MA, USA) according to the manufacturer’s protocol. The U87, U118, and Hs5 cells (5 × 10^4^ cells per well) were seeded onto 6-well plates and incubated overnight. The next day, the medium was replaced with a fresh medium containing NDs at concentrations 5 or 50 μg/mL, and the cells were cultivated for 24 or 72 h. Then, the cells were washed in PBS and stained using the kit. The BD FACSCalibur™ cytometer (Becton Dickinson, Franklin Lakes, NJ, USA) was used to measure the fluorescence emission at 530 and 575 nm using excitation at 488 nm. Annexin V staining was detected as green fluorescence and PI as red fluorescence.

### 3.8. Doubling Time Assay

To estimate the time required by cells to duplicate their number, doubling time assays were performed. A total of 1 × 10^4^ U87 and U118 cells were seeded onto 6-well tissue culture plates. Following overnight incubation, fresh medium with ND at concentrations 5 or 50 μg/mL was introduced to the cells. After 24 or 72 h, the cells were washed with 2 mL/well PBS, harvested with trypsin, re-suspended in 1 mL/sample in complete culture medium, and finally counted in a TC10™ automated cell counter (BioRad, Hercules, CA, USA). The doubling time was calculated using the doubling time online calculator [43].

### 3.9. Cell Proliferation Assay

Cell proliferation was evaluated using the bromodeoxyuridine (BrdU) incorporation assay (Cell Proliferation ELISA BrdU kit; Thermo Fisher Scientific, MA, USA). The U87, U118 and Hs5 cells (5 × 10^3^ cells per well) were seed onto 96-well plates and incubated overnight. Then, they were treated with ND at concentrations 5 or 50 μg/mL and incubated for 24 or 72 h. The BrdU assay was performed according to the manufactures’ protocol. The absorbance was measured at 450 nm using a plate reader (Infinite M200, Tecan, Durham, NC, USA). Cell proliferation is expressed as a ratio of the relative optical density of tested samples (*OD_test_*
− *OD_blank_*) to the relative optical density of the control sample (*OD_control_*
− *OD_blank_*); both relative values were calculated with respect to the optical density of the blank.

### 3.10. c.DNA Synthesis and qPCR Analysis

Total RNAs were isolated from the same number of U87 and U118 cells treated with 50 μg/mL ND for 24 and 72 h, according to the kit manufacturer’s instructions (Blood/Cell RNA Mini Kit; Syngen Biotech, Wroclaw, Poland). First-strand cDNAs were synthesized from 0.7 μL of the total isolated RNA using the High-Capacity cDNA Reverse Transcription Kit (Thermo Fisher Scientific, MA, USA). The qPCR analyses were performed by Power SYBR Green Master Mix (Thermo Scientific). The reactions were conducted with a total volume of 10 μL in StepOne Real-Time PCR System (Applied Biosystems Inc., Carlsbad, CA, USA).

Gene-specific primers (Table 2) were purchased from Genomed (Warsaw, Poland), and *TBP* was used as the reference gene [44]. The amplification was carried out as follows: reaction runs for 2 min at 50 °C and 10 min at 95 °C, followed by 40 cycles of a two-step PCR consisting of a denaturing phase at 95 °C for 15 s, and a combined annealing and extension phase at 72 °C for 30 s. Relative gene expression (RQ) was calculated from the formula 2^−ΔΔCT^, where ΔΔCT = ΔCT of a control − ΔCT of a treated sample and ΔCT = mean CT of *TBP*
− CT of a target gene.

### 3.11. Cell–Cycle Analysis

Changes in the cell cycle were analyzed by measuring the proportion of cells undergoing different phases using propidium iodide DNA staining and flow cytometry. The U87 and U118 cells (5 × 10^4^ cells per well) were seeded onto six-well plates and incubated overnight. Afterwards, the cells were treated with 5 or 50 μg/mL ND for 24 or 72 h, fixed in ice-cold 70% ethanol, and then stored at 4 °C until PI staining. The ethanol-suspended cells were centrifuged at 1000 rpm for 5 min and washed twice in PBS. The pellets were suspended in 1 mL of PI/RNase A reagent and incubated at 37 °C for 30 min. The cell cycle profiles were obtained using a FACS Cell Flow Cytometer (Becton Dickinson USA). Noise in the data was gated out during data acquisition and 10,000 events were collected from each determination. Data were analyzed with the Cell Quest Pro software. All measurements were performed in triplicates.

### 3.12. Statistical Analysis

Data were analyzed using one-way analysis of variance (ANOVA) and the Bonferroni post-hoc test in comparison to the control values. Results are shown as mean values with standard deviations. Differences at *p* < 0.05 were considered significant. All data were analyzed using GraphPad Prism 7 (GraphPad Software Inc., La Jolla, CA, USA).

## 4. Conclusions

In conclusion, all cancer cells have identical features, i.e., avoidance of cell death and intense proliferation. The unlimited cell division in cancer cells is mainly due to aberrant cell cycle progression. Most of the tumor therapies focus on the activation of cell apoptosis; however, as a consequence, cell proliferation increases, eventually defending the tumor against destruction [45]. Therefore, simultaneous activation of apoptosis and inhibition of cell proliferation is extremely important as it allows restoration of the proper balance between life and death. We have previously shown that NDs have anti-angiogenic properties [18] and can inhibit cell migration and adhesion [20]. In the present study, we have also demonstrated their anti-proliferative properties exerted through the G_0_/G_1_-phase arrest of the cell cycle, hence indicating the potential of NDs in glioblastoma therapy.

## Figures and Tables

**Figure 1 molecules-24-01549-f001:**
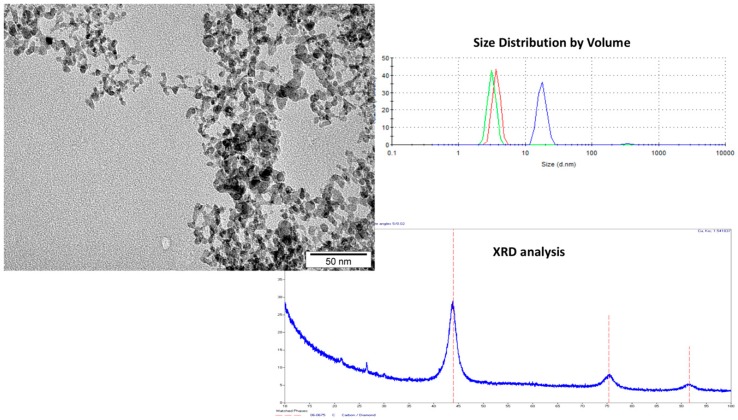
Physicochemical analyses (TEM, DLS, XRD) of diamond nanoparticles (NDs). Scale bar represents 50 nm. TEM, transmission electron microscopy; DLS, dynamic light scattering; XRD, X-ray diffraction.

**Figure 2 molecules-24-01549-f002:**
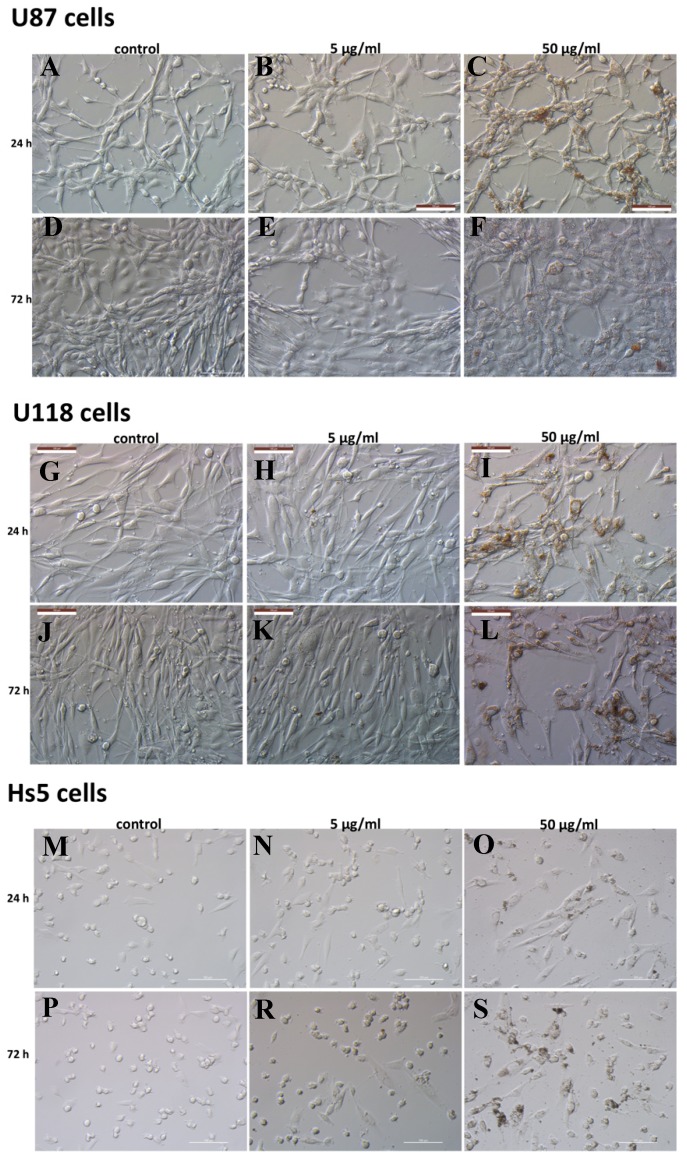
Morphological changes in U87 (**A**–**F**), U118 (**G**–**L**) glioblastoma cells, and Hs5 stromal cells (**M–S**) after treatment with 5 and 50 μg/mL concentrations of diamond nanoparticles (NDs). Represents control cells (**A**,**D**; **G**,**J**; **M**,**P**) that did not undergo treatment. Scale bar represents 100 μm.

**Figure 3 molecules-24-01549-f003:**
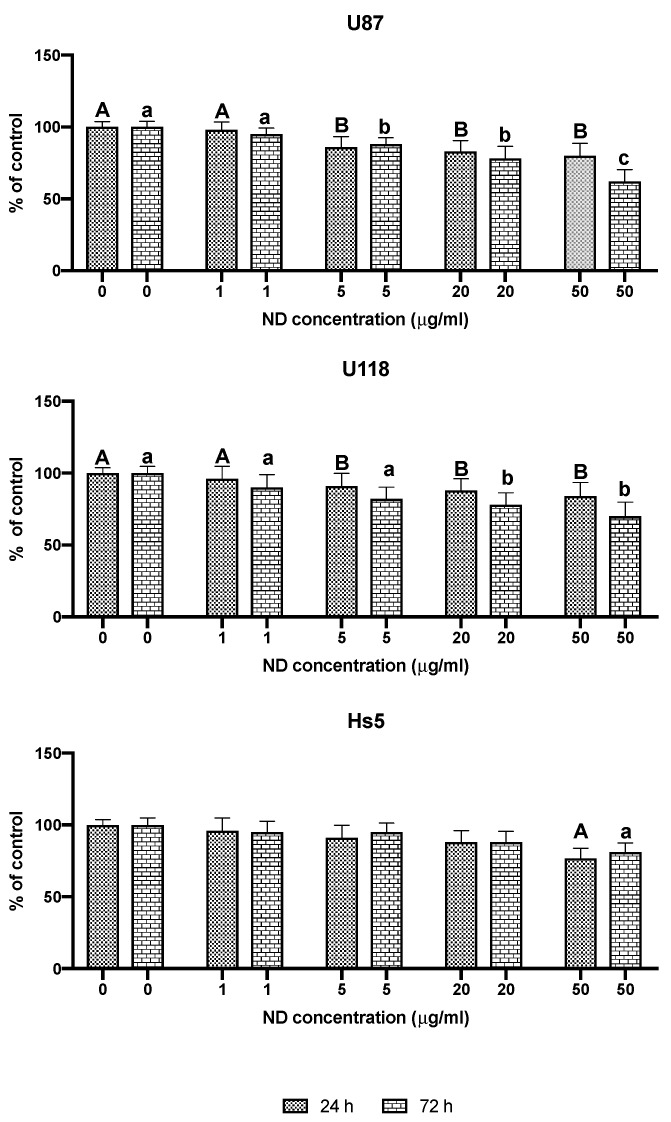
Viability of Hs5, U87, and U118 cells after treatment with nanoparticles of diamond (NDs) at concentrations of 1, 5, 20, and 50 μg/mL for 24 and 72 h. Groups marked with different letters (A, B for 24 h treatment and a, b, c for 72 h treatment) are significantly different (*p* < 0.05).

**Figure 4 molecules-24-01549-f004:**
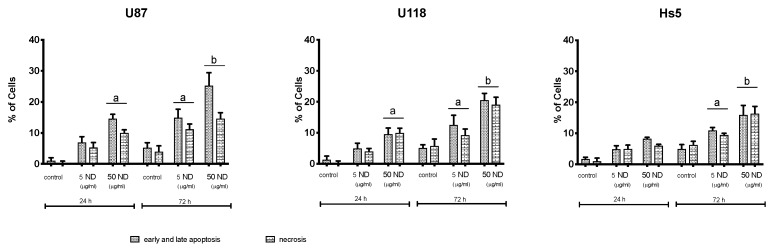
Analysis of apoptosis in U87, U118 and Hs5 cell lines after diamond nanoparticles (NDs) treatment at concentrations of 5 and 50 μg/mL for 24 and 72 h. The indicated values represent the measurements from the Annexin V-Alexa Fluor^®^ 488 and propidium iodide assay analyses. After the 50 μg/mL ND treatment and incubation for 24 and 72 h, the frequency of cell death (both early and late apoptosis and necrosis) increased (*p* < 0.05).

**Figure 5 molecules-24-01549-f005:**
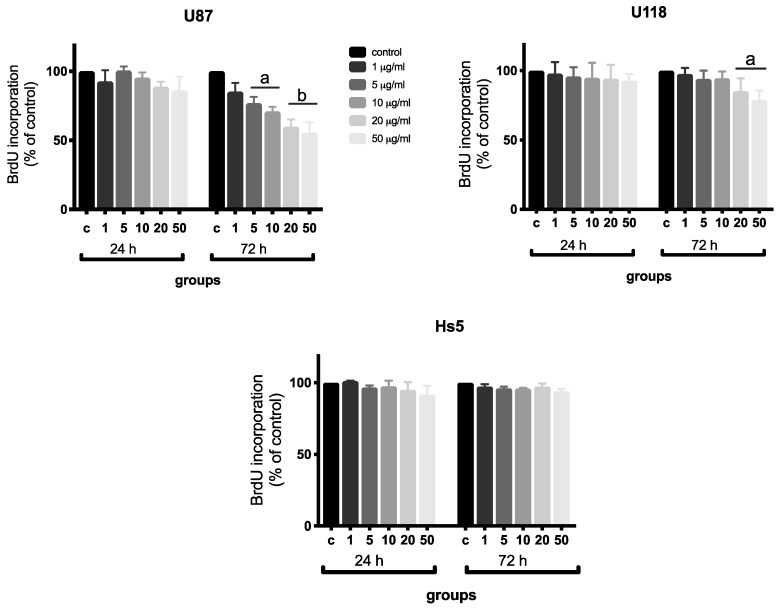
Analysis of proliferation of the Hs5, U87, and U118 cell lines after diamond nanoparticle (ND) treatment at concentrations of 1, 5, 10, 20, and 50 μg/mL for 24 and 72 h using the Bromodeoxyuridine (BrdU) assay. Values are represented as mean ± standard deviation of the relative percentage of cells to the BrdU positive cells. Groups marked with different letters (a, b) are significantly different (*p* < 0.05).

**Figure 6 molecules-24-01549-f006:**
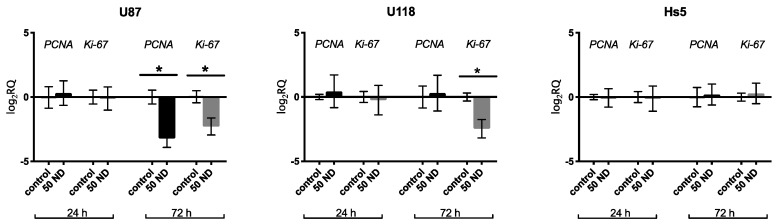
Analysis of the mRNA expression level of *PCNA* and *Ki-67* after ND treatment at 50 μg/mL concentration for 24 and 72 h in U87, U118, and Hs5 cells. Groups marked with an asterisk (*) are significantly different (*p* < 0.05). The results are calculated relative to the control values. Log_2_RQ values for all genes are normalized to the *TBP* housekeeping gene. *PCNA*, proliferating cell nuclear antigen; *TBP*, TATA box binding protein.

**Figure 7 molecules-24-01549-f007:**
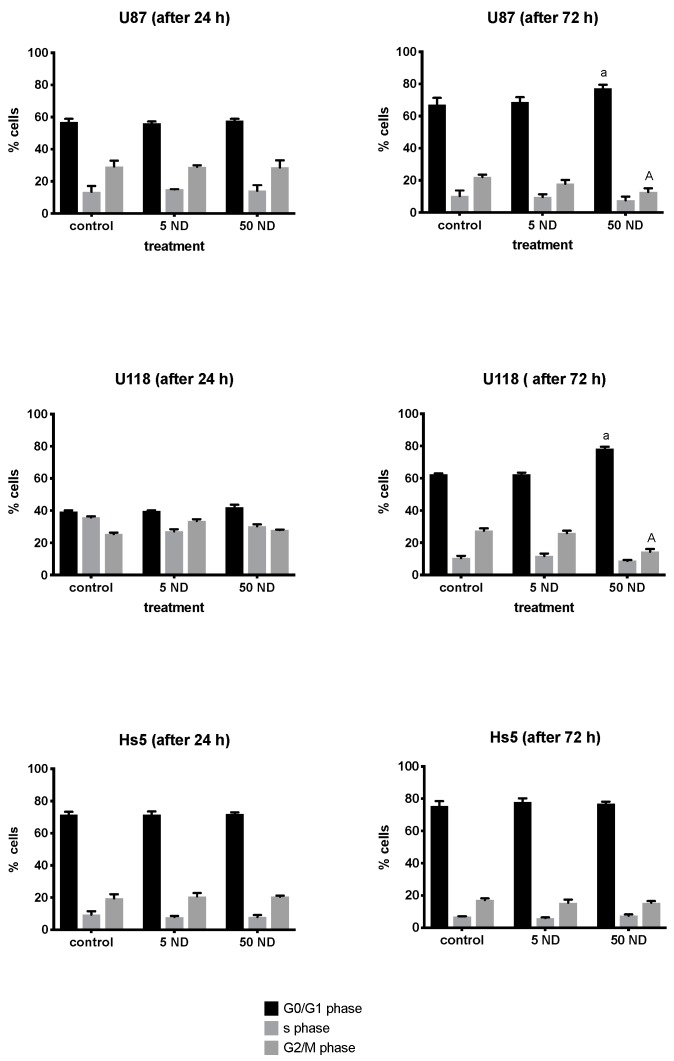
Flow cytometric analysis of Hs5, U87, and U118 cell cycle ND treatment at concentrations of 5 and 50 μg/mL for 24 and 72 h. Values represent the relative percentage of cells at the G0/G1, S, and G2/M phases of the cell cycle. Groups marked with different letters (a, A) are significantly different (*p* < 0.05). G_0_, resting phase (outside of replication); G_1,_ interval between mitosis and initiation of DNA replication; S, phase of DNA replication; G_2_, interval between DNA replication and mitosis; M, Mitosis.

**Figure 8 molecules-24-01549-f008:**
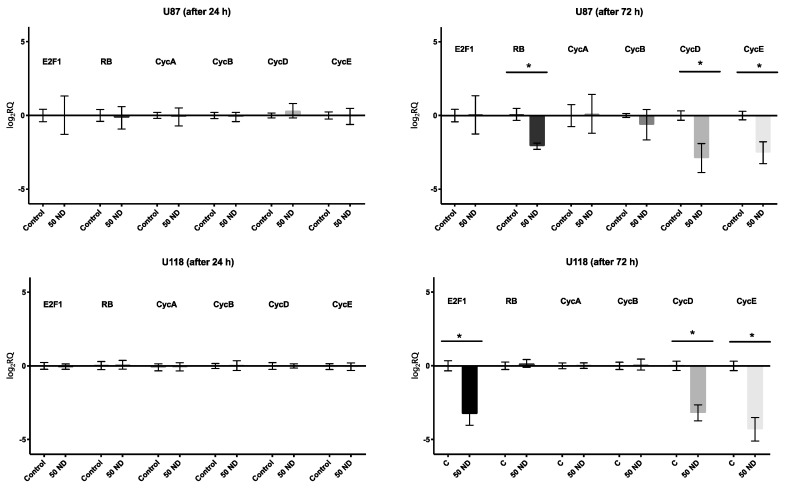
Analysis of the mRNA expression levels of *E2F1*, *Rb*, *CycA*, *CycB*, *CycD*, *CycE*. The U87 and U118 cells were treated with 50 μg/mL diamond nanoparticles (NDs) for 24 and 72 h. Groups marked with an asterisk (*) are significantly different (*p* < 0.05). The results are calculated relative to the control values. Log_2_RQ values for all genes are normalized to the *TBP* housekeeping gene. *TBP*, TATA-box binding protein; *CycA*, Cyclin A2; *CycD*, Cyclin D2; *CycB*, Cyclin B1; *CycE*, Cyclin E2; *Rb*, transcriptional corepressor 1; *E2F1*, E2F transcription factor 1; *PCNA*, proliferating cell nuclear antigen; *Ki-67*, marker of proliferation KI-67.

**Figure 9 molecules-24-01549-f009:**
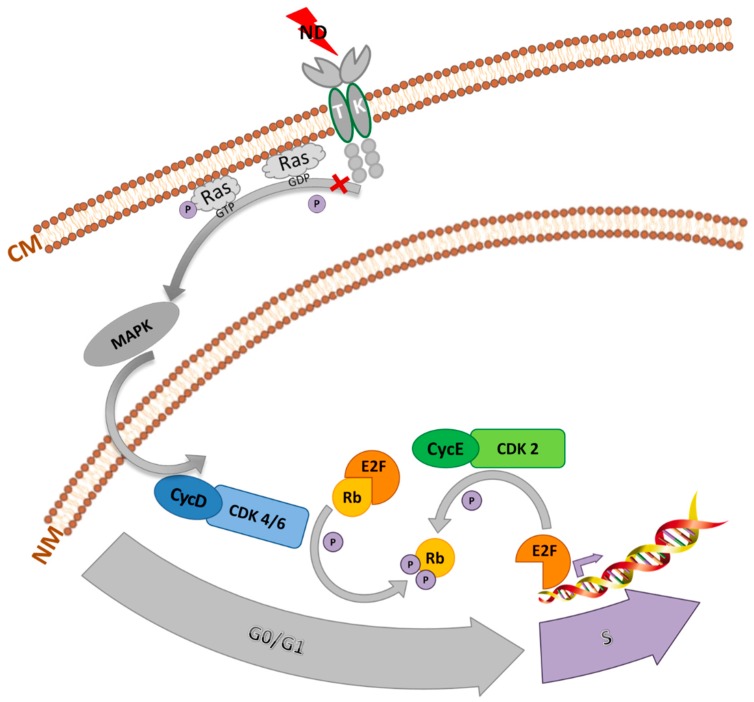
Proposed mechanism of antiproliferative action of diamond nanoparticles.

**Table 1 molecules-24-01549-t001:** Doubling time of U87, U118, and Hs5 cell lines after diamond nanoparticle (ND) treatment at concentrations of 5 and 50 μg/mL for 24 and 72 h.

	U87			U118			Hs5		
	Control	5 μg/mL ND	50 μg/mL ND	Control	5 μg/mL ND	50 μg/mL ND	Control	5 μg/mL ND	50 μg/mL ND
Incubation time, h									
0	3.0 × 10^4^	3.0 × 10^4^	3.0 × 10^4^	3.00 × 10^4^	3.00 × 10^4^	3.00 × 10^4^	3.00 × 10^4^	3.0 × 10^4^	3.0 × 10^4^
24	5.2 × 10^4^	5.0 × 10^4^	3.5 × 10^4^	6.92 × 10^4^	6.89 × 10^4^	6.60 × 10^4^	3.80 × 10^4^	3.80 × 10^4^	3.80 × 10^4^
72	10^5^	9.2 × 10^4^	6.7 × 10^4^	3.90 × 10^5^	4.05 × 10^5^	3.00 × 10^5^	6.05 × 10^4^	6.00 × 10^4^	6.00 × 10^4^
Doubling time, h	32	34	39	19	19	23	71	72	72

**Table 2 molecules-24-01549-t002:** Sequence of primers used in the qPCR analysis.

Gene	Sequence	Source
*TBP*	F: GAGCTGTGATGTGAAGTTTCC R: TCTGGGTTTGATCATTCTGTAG	44
*CycA2*	F: TTATTGCTGGAGCTGCCTTT R: CTCTGGTGGGTTGAGGAGAG	45
*CycD2*	F: TACTTCAAGTGCGTGCAGAAGGAC R: TCCCACACTTCCAGTTGCGATCAT	sp
*CycB1*	F: GGCTTCCTCTTCACCAGGCA R: CGCGATCGCCCTGGAAAC	sp
CycE2	F: AATCAGGCAAAGGTGAAGGA R: CCCCAAGAAGCCCAGATAAT	sp
*Rb1*	F: CGGGAGTCGGGAGAGGACGG R: CGAGAGGCAGGTCCTCCGGG	sp
*E2F1*	F: ACCTTCGTAGCATTGCAGACC R: TTCTTGCTCCAGGCTGAGTAG	46
*PCNA*	F: CCATCCTCAAGAAGGTGTTGG R: GTGTCCCATATCCGCAATTTTAT	47
*Ki-67*	F: CCACACTGTGTCGTCGTTTG R: CCGTGCGCTTATCCATTCA	sp

F, forward, sequence 5′->3′; R, reversed, sequence 3′->5′; sp, self-projected; *TBP*, TATA-box binding protein; *Cyc-A2*, Cyclin A2; *Cyc-D2*, Cyclin D2; *Cyc-B1*, Cyclin B1; *Cyc-E2*, Cyclin E2; *Rb1*, Transcriptional corepressor 1; *E2F1*, E2F transcription factor 1; *PCNA*, Proliferating cell nuclear antigen; *Ki-67*, Marker of proliferation Ki-67.

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
