# Peer review of "Diamond Nanoparticles Downregulate Expression of CycD and CycE in Glioma Cells"

_molecules, 2019, doi:10.3390/molecules24081549_

Reviewer 1 Report

The manuscript titled “Diamond nanoparticles down-regulate expression of CyD and CycE in glioma cells” by Marta Grodzik  et. al., demonstrates that diamond nanoparticles exhibite proliferation inhibition and altered cell cycle in two fast-dividing glioma cell lines. The authors include different types of analysis including cell morphology, viability, cell cycle and double time assays and different gen expressions that have been selected to further explore these findings. The topic of the medical use of diamond nanoparticles (ND) for medical use is important from several perspectives and uses and has recently gained interest in the nanomedicine scientific community due to their low cytotoxicity and high biocompatibility. Authors rationally summarize the proliferation results including the expression of two genes involved in the regulation of cell proliferation with elongated doubling time and decreased expression in cancer cell lines (similarly to the results found in article 13 by the same authors). Similarly, the analysis of cell cycle progression by flow cytometric analysis of control and cancer cells at different cell phases and, mainly by the differential mRNA expression using qPCR of key regulator proteins of this pathway seems to support the conclusion of an antiproliferative mechanism of ND with cell cycle stop between mitosis and the replication of DNA as the authors claim. The article is well written and can be potentially useful to researchers working in this field and for the normal reader of this journal, however I found that main problem is its very limited scope. Authors are just presenting results based on morphology changes and qPCRs. The authors should add additional and key data, e.g. caspase-mediated apoptosis and mitochondrial membrane potential to reflect complementary aspects of cellular viability and physiology, essential to explain their results. Additionally, since authors are using commercially available nanoparticles. I presume that the curious reader that do not know anything about this, especially those working in biological applications will need to know further details of the composition or the coating of these nanoparticles, not only the size and shape characterization. Results and interpretations will be completely different without a clear identification of these factors. 

In my opinion, this work will enhance interest of the scientific community of the nanomaterials and fits well to the scope of the journal, but adding much more information is needed. The manuscript is clear, well organized, and well written however I have raised this major concern regarding its potential applicability as a therapeutic agent in tumor or highly dividing cells as these authors are claiming.

 Author Response

We would like to thank the Reviewer for taking time to review our manuscript. We agree with his comments and we improved the manuscript accordingly. We hope he will find this revision rised to his expectations.

All changes are in red.

Reviewer [R]: The manuscript titled “Diamond nanoparticles down-regulate expression of CyD and CycE in glioma cells” by Marta Grodzik  et. al., demonstrates that diamond nanoparticles exhibite proliferation inhibition and altered cell cycle in two fast-dividing glioma cell lines. The authors include different types of analysis including cell morphology, viability, cell cycle and double time assays and different gen expressions that have been selected to further explore these findings. The topic of the medical use of diamond nanoparticles (ND) for medical use is important from several perspectives and uses and has recently gained interest in the nanomedicine scientific community due to their low cytotoxicity and high biocompatibility. Authors rationally summarize the proliferation results including the expression of two genes involved in the regulation of cell proliferation with elongated doubling time and decreased expression in cancer cell lines (similarly to the results found in article 13 by the same authors). Similarly, the analysis of cell cycle progression by flow cytometric analysis of control and cancer cells at different cell phases and, mainly by the differential mRNA expression using qPCR of key regulator proteins of this pathway seems to support the conclusion of an antiproliferative mechanism of ND with cell cycle stop between mitosis and the replication of DNA as the authors claim. The article is well written and can be potentially useful to researchers working in this field and for the normal reader of this journal, however I found that main problem is its very limited scope. Authors are just presenting results based on morphology changes and qPCRs. The authors should add additional and key data, e.g. caspase-mediated apoptosis and mitochondrial membrane potential to reflect complementary aspects of cellular viability and physiology, essential to explain their results.

Answer [A]: This article is our next one about Diamond Nanoparticles (DN) and their influence on glioma cells. We also published results on DN effect on chicken embryo development, toxicity in rats and angiogenesis on Cholioallanatoic membrane (CAM) model. In addition to these articles we are preparing new manuscript about the influence of ND on cell death and aging in glioma (not only apoptosis but also autophagy, necrosis and necroptosis). This is the reason why in the current article the subject is rather narrowly presented. Other issues were or will be presented in other manuscripts. However, we understand Reviewer’s objections. We added to the Manuscript results of analysis of apoptosis/necrosis by PI/Anexin V assay. 

The following is added/modified:

Line 133-144: The percentage of apoptotic cells (early and late in total) (Figure 4) increased to 14.5 and 25.2% after treating the U87 cells with 50μg/mlof ND for 24 and 72 h, respectively; in the case of U118 cells, the percent increased to 9.5 and 20.6% after 24 and 72 h, respectively. These results demonstrate the ability of ND to induce apoptosis in U87 and U118 cells in a time-dependent manner. In Hs5 cell apoptosis/necrosis were induced with 50μg/mlof ND for 72 h, however the percentage of death cells was lower than in U87 and U118 group.

Figure 4.Analysis of apoptosis in U87 and U118 cell lines after diamond nanoparticles (ND) treatment at concentrations of 5 and 50 μg/ml for 24 and 72 h. The indicated values represent the measurements from the Annexin V-Alexa Fluor®488 and propidium iodide assay analyses. After the 50 μg/ml ND treatment and incubation for 24 and 72 h, the frequency of cell death (both early and late apoptosis and necrosis) increased (P<0.05).< span="">

Line 321-330: 3.7. Apoptosis and Necrosis Assay

Annexin V and propidium iodide (PI) staining for the apoptosis/necrosis assaywas performed using the Alexa Fluor®488 Annexin V/Dead Cell Apoptosis Kit (Thermo Fisher Scientific) according to the manufacturer’s protocol. The U87, U118 and Hs5 cells (5x104cells per well) were seeded onto 6-well plates and incubated overnight. The next day, the medium was replaced with a fresh medium containing ND at concentrations 5 or 50 μg/mL, and the cells were cultivated for 24 or 72 h. Then, the cells were washed in PBS and stained using the kit. TheBD FACSCalibur™ cytometer (Becton Dickinson, Franklin Lakes, NJ, USA) was used to measure the fluorescence emission at 530 and 575 nm using excitation at 488 nm. Annexin V staining was detected as green fluorescence and PI as red fluorescence. 

[R]: Additionally, since authors are using commercially available nanoparticles. I presume that the curious reader that do not know anything about this, especially those working in biological applications will need to know further details of the composition or the coating of these nanoparticles, not only the size and shape characterization. Results and interpretations will be completely different without a clear identification of these factors. 

[A]: We agree with the Reviewer. Description of the surface characteristics has been added.

 [R]: In my opinion, this work will enhance interest of the scientific community of the nanomaterials and fits well to the scope of the journal, but adding much more information is needed. The manuscript is clear, well organized, and well written however I have raised this major concern regarding its potential applicability as a therapeutic agent in tumor or highly dividing cells as these authors are claiming.

 [A]: Thank you very much for kind words. We appreciate them.

 We noticed a small inacuracy in Figure 9 so we improved it

Line 275:

 Reviewer 2 Report

This manuscript describes a detailed exploration into the use of nanodiamonds in the treatment of aggressive neoplastic disease, in this case glioblastoma multiforme. Various cell lines have been challenged with nanodiamonds to demonstrate their potential and there is an elegant study to demonstrate the point(s) within the cell cycle that may be inhibited or arrested by the presence of nanodiamonds. Overall, this is a good and well presented piece of research and the manuscript provides a lot of data to support the the findings.

I have a few comments:

The introduction contains quite a few abbreviations and acronyms. Make sure that each is explained fully in the text, e.g. GBM is defined in the abstract but not the introduction.

The introduction also highlights other biomedical investigations using nanodiamonds, indicating the number of groups interested in this field and the timeliness of the work. However, the manuscript must contain some comparison of nanodiamonds with other nanoparticle systems for the treatment of cancers. Such information is contained within the following review

Ozdemir-Kaynak E, Qutub AA and Yesil-Celiktas O (2018) Advances in Glioblastoma Multiforme Treatment: New Models for Nanoparticle Therapy. Front. Physiol. 9:170. doi: 10.3389/fphys.2018.00170

and other papers which are available widely.

Similarly, mention must be made about the possible fate of nanodiamonds in vivo, drawing on any evidence base. There is much interest in the fate of nanoparticles in vivo and the potentially toxic effects they may have beyond the beneficial effects for treatment of cancers, for example. This does not have to be extensive but the authors must be mindful of this.

Author Response

We would like to thank the Reviewer for his thoughtful review of the manuscript. All the comments and suggestions have been considered.

Reviewer (R]: The introduction contains quite a few abbreviations and acronyms. Make sure that each is explained fully in the text, e.g. GBM is defined in the abstract but not the introduction.

We agree with the Reviewer. The following explanation has been added/modified:

Line 43: Glioblastoma multiforme (GBM)

(R]: The introduction also highlights other biomedical investigations using nanodiamonds, indicating the number of groups interested in this field and the timeliness of the work. However, the manuscript must contain some comparison of nanodiamonds with other nanoparticle systems for the treatment of cancers. Such information is contained within the following review Ozdemir-Kaynak E, Qutub AA and Yesil-Celiktas O (2018) Advances in Glioblastoma Multiforme Treatment: New Models for Nanoparticle Therapy. Front. Physiol. 9:170. doi: 10.3389/fphys.2018.00170 and other papers which are available widely.

We agree with the Reviewer. The following section has been added/modified:

Line 47-54: The current standard therapy is maximal surgical resection followed by radiotherapy and chemotherapy with temozolamide [4]. The main problem in chemotherapy is presence of the blood-brain barrier (BBB) which controlled blocks toxins pass as well as many essential drugs from reaching brain tissue. Nanotechnology and mathematical modeling nanoparticles can help delivery of active compounds (for example delphinidin) to brain tissue. [5] Another authors used polymeric nanoparticle with Paclitaxel [6], polymeric micelles with doxorubicin [7] and liposomes with siRNA [8] for convection-enhanced delivery. In the current article we focus on nanoparticle not as a vehicle for delivery but as an active agent with anticancer properties.  

(R]: Similarly, mention must be made about the possible fate of nanodiamonds in vivo, drawing on any evidence base. There is much interest in the fate of nanoparticles in vivo and the potentially toxic effects they may have beyond the beneficial effects for treatment of cancers, for example. This does not have to be extensive but the authors must be mindful of this.

We agree with the Reviewer. The following information has been added/modified:

Line 60-63: ND were also tested for their application in cancer chemiotherapy. Yu et al [16] synthesied diamond-based nanoparticles which can penetrate cell membrane in vitro and in vivo.On the other hand ND functionalized by Epirubicin improved impairment of secondary tumor formation in vivo [17].

We noticed a small inacuracy in Figure 9 so we improved it

Line 275:

Round  2

Reviewer 1 Report

no further comment. The authors answered and included all the work and results that were claimed in my previous review.